# Assessing the Accuracy and Reliability of the Monitored Augmented Rehabilitation System for Measuring Shoulder and Elbow Range of Motion

**DOI:** 10.3390/s25144269

**Published:** 2025-07-09

**Authors:** Samuel T. Lauman, Lindsey J. Patton, Pauline Chen, Shreya Ravi, Stephen J. Kimatian, Sarah E. Rebstock

**Affiliations:** 1Department of Anesthesiology and Pain Management, University of Texas Southwestern, Dallas, TX 75390, USA; stephen.kimatian@utsouthwestern.edu (S.J.K.); sarah.rebstock@utsouthwestern.edu (S.E.R.); 2Children’s Health, 1935 Medical District Drive, Dallas, TX 75235, USA; lindsey.patton@childrens.com (L.J.P.); peng.chen@childrens.com (P.C.); shreya.ravi@childrens.com (S.R.)

**Keywords:** biomechanics, range of motion, motion analysis, accuracy, reliability, markerless motion tracking

## Abstract

Accurate range of motion (ROM) assessment is essential for evaluating musculoskeletal function and guiding rehabilitation, particularly in pediatric populations. Traditional methods, such as optical motion capture and handheld goniometry, are often limited by cost, accessibility, and inter-rater variability. This study evaluated the feasibility and accuracy of the Microsoft Azure Kinect-powered Monitored Augmented Rehabilitation System (MARS) compared to Kinovea. Sixty-five pediatric participants (ages 5–18) performed standardized shoulder and elbow movements in the frontal and sagittal planes. ROM data were recorded using MARS and compared to Kinovea. Measurement reliability was evaluated using intraclass correlation coefficients (ICC3k), and accuracy was evaluated using root mean squared error (RMSE) analysis. MARS demonstrated excellent reliability with an average ICC3k of 0.993 and met the predefined accuracy threshold (RMSE ≤ 8°) for most movements, with the exception of sagittal elbow flexion. These findings suggest that MARS is a reliable, accurate, and cost-effective alternative for clinical ROM assessment, offering a markerless solution that enhances measurement precision and accessibility in pediatric rehabilitation. Future studies should enhance accuracy in sagittal plane movements and further validate MARS against gold-standard systems.

## 1. Introduction

Range of motion (ROM) measurement is essential for assessing musculoskeletal function, providing insights into joint health, flexibility, and overall mobility. Accurate ROM measurements allow clinicians to track progress, identify impairments, and develop individualized rehabilitation programs aimed at improving functional outcomes [1]. This is particularly vital in pediatric populations, where precise ROM evaluation is essential for quantifying functional capacity and supports the development of skills necessary for activities of daily living (ADLs) [2,3].

Optical motion capture systems are widely regarded as the gold standard for joint ROM assessment due to their exceptional accuracy and reproducibility. However, their high cost, lack of portability, and requirement for technical expertise limit their practicality in routine clinical environments. As a result, alternative methods such as Kinovea, a free, open-source two-dimensional (2D) video analysis software, have gained attention. Kinovea has demonstrated subpixel measurement accuracy [4], with studies reporting strong reliability for ROM assessment when compared to motion capture systems. For example, Fernández-González, et al. [5] found intra-rater reliability of ±2.5° and inter-rater of ±5° when comparing lower extremity ROM to a Vicon motion capture system. While Elrahim, et al. [6] concluded that Kinovea is a reliable tool for measuring shoulder ROM, however, it is yet to be seen if this accuracy translates to pediatric populations. Despite these advantages, Kinovea has notable limitations, including its reliance on manual annotation and its constraint to 2D analysis, which may limit precision and increase user-dependent variability. Nonetheless, its accessibility, prior validation, and existing clinical use make it a pragmatic comparator for evaluating new technologies such as MARS, particularly within pediatric settings where cost and feasibility are key concerns.

Handheld goniometry has historically been the primary tool for ROM measurement in clinical settings due to its simplicity, low cost, and portability. In recent years, digital goniometry such as systems based on inertial measurement units (IMUs), has gained popularity in both research and commercial contexts, with several studies validating their accuracy and clinical utility [7,8]. However, these systems often require calibration, user training, and wearable sensors, and they may carry higher upfront and maintenance costs compared to traditional tools. Importantly, their reliance on proper sensor placement and participant compliance makes them less suitable for pediatric populations, where movement unpredictability and limited tolerance for body-worn devices pose practical barriers. In contrast, markerless systems such as MARS offer contact-free, automated ROM assessment that is inherently more adaptable to pediatric needs.

The Microsoft Azure Kinect camera system, a depth sensor-based device with body-mapping capabilities, has gained popularity for its use in motion, gait, and posture analysis. When integrated with the Monitored Augmented Rehabilitation System (MARS) software, this system offers a promising, low-cost, markerless solution for ROM assessment. Such a system holds considerable potential in clinical environments, where traditional ROM measurement methods can be cost-prohibitive, inaccessible, or inefficient.

Despite the availability of validated tools like Kinovea, no markerless system to date has been clinically validated for pediatric ROM assessment that combines real-time measurement, task-based function, and low-cost implementation. This study aimed to evaluate the feasibility and accuracy of the Microsoft Azure Kinect camera system, integrated with MARS software. Its performance was compared to the widely used Kinovea software for ROM assessment in pediatric clinical settings. We hypothesized that MARS would demonstrate excellent reliability (ICC ≥ 0.9) and achieve a root mean squared error (RMSE) of ≤8° across shoulder and elbow movements in the frontal and sagittal planes. Additionally, this study sought to validate MARS as a viable clinical tool in pediatric clinical settings, while identifying areas for refinement and further development.

## 2. Materials and Methods

### 2.1. Participant Demographics

Sixty-five children and adolescents (29 male; 36 female) aged 5 to 18 years (mean age: 12.71 ± 3.3 years) participated in this study. The average height of the participants was 155 ± 15.9 cm, and the mean weight was 53 ± 20.2 kg, resulting in an average body mass index (BMI) of 21.67 ± 6.2. Subjects were excluded from participation if they had any musculoskeletal disease or disorder, developmental delay, or any other condition that would prohibit completion of the required movements.

This age range was selected to capture a representative spectrum of pediatric musculoskeletal development as encountered in typical outpatient rehabilitation settings. While this introduces natural variation in joint mobility and movement control, the intent was to evaluate MARS performance across a clinically relevant pediatric population. All participant demographic information is shown in Table 1.

### 2.2. Sample Size Determination

This study performed a simulation analysis to determine the ideal sample size for achieving stable RMSE values. The RMSE values were assessed across sample sizes of 10, 15, 20, 25, 30, 35, and 38 participants. The analysis showed that while RMSE values fluctuated significantly with smaller samples, they became consistent and reliable with 38 participants. These results suggest that a sample size of 38 participants is sufficient to achieve stable and reliable RMSE estimates.

### 2.3. Ethical Aspects

All participants and their parents read and signed an informed consent approved by the University of Texas Southwestern Institutional Review Board (STU-2020-1401).

### 2.4. Instrumentation

ROM data were captured using the Microsoft Azure Kinect camera system (Microsoft Inc., Redmond, WA, USA), which utilizes a 12 MP depth camera and 12MP CMOS sensor rolling shutter red–green–blue (RGB) camera. The depth camera operates at 1024 × 1024 pixels, and the RGB camera provides a resolution of 3840 × 2160 pixels. Data were collected at a frame rate of 30 Hz for both RGB and depth streams to ensure synchronization with joint tracking data. One Azure Kinect camera was positioned 2.3 m in front of the participant to capture frontal and sagittal plane movements, while a second camera, placed 2.3 m to the left, was used to validate sagittal plane motion. A schematic of the setup is shown in Figure 1.

Joint ROM data were analyzed using MARS (Version 0.6), a Windows application built with the Unity game engine that integrates the Azure Kinect Sensor SDK and the Azure Kinect Body Tracking SDK. The Azure Kinect captured depth and image data, which were processed by the Azure Kinect Body Tracking SDK using a convolutional neural network (CNN) for 2D joint estimation and body segmentation. Joint angles for shoulder flexion/extension, shoulder abduction/adduction, and elbow flexion/extension were calculated in MARS once per body frame based on the tracked skeletal data. For each angle of interest, two vectors were formed from three anatomical landmarks.

### 2.5. Procedures

Each participant performed three rounds of movement sequences, each lasting 40 s, aimed at assessing ROM for the shoulder and elbow joint in the frontal and sagittal planes. Patients were instructed to conduct their movements in a controlled manner as they followed the direction of the research staff. Movements included full shoulder abduction and elbow flexion in the frontal plane, and full shoulder flexion and elbow flexion in the sagittal plane.

During the three rounds of movement sequences, measurements were taken at specific joint angles to evaluate ROM. For example, in the case of shoulder abduction, measurements were recorded at 45°, 90°, and 135° during round 1, offset by −15° in round 2 (30°, 75°, and 120°), and offset by +15° in round 3 (60°, 105°, and 150°). This protocol allowed for a comprehensive assessment across a wider ROM whilst accounting for variability in movement patterns.

Although neutral position (0°) is conventionally used as a biomechanical baseline, we observed variability in participants’ relaxed arm postures during pilot testing, likely due to their age and skeletal maturity. To minimize the influence of structural differences at rest and to improve consistency, we selected 45° as the initial starting angle for ROM stratification. This threshold ensured all measured movements began from a functionally equivalent position, enhancing the internal consistency of joint angle comparisons.

Joint angle calculations were derived using the Azure Kinect Body Tracking SDK, which applies a convolutional neural network to estimate 3D positions of 32 anatomical landmarks based on depth and RGB input. For the purposes of this study, joint angles were calculated in MARS by defining vectors between SDK-tracked joint positions corresponding to standardized anatomical references utilizing shoulder, elbow, and wrist markers.

Frontal plane movements were analyzed using the video output of the front camera using the joint angle calculation feature of Kinovea (Version 0.9.5. Charmant, J. and contributors, http://www.kinovea.org). While sagittal plane movements were ascertained using the angles reported by the front camera, before being validated by Kinovea using the video obtained from the second, side camera (Figure 1 and Figure 2). Elbow ROM measures were isolated and used the center point of the elbow as the axis, with measurement arms positioned on the wrist and shoulder markers (Figure 3).

To maximize accuracy for shoulder ROM (flexion and abduction), postural compensation of the torso was incorporated to account for any deviation of an individual’s torso to one side or another. This compensation was applied post hoc using Kinovea and measured by calculating the angle between the mid-pelvic marker to the neck marker in the frontal plane, and the lateral view pelvic marker to the neck marker in the sagittal plane (Figure 2). Shoulder movements were subsequently calculated using the vertical angle calculation and placing markers on the shoulder and elbow joints.

Each measure collected by MARS was recorded and compared to the Kinovea calculation for accuracy. Kinovea was selected for comparison due to its established use in clinical and research settings as a validated alternative to high-cost motion capture systems. Its subpixel measurement accuracy and strong reliability in previous studies make it a practical benchmark for evaluating MARS’ accuracy in pediatric populations.

Prior to analysis, all MARS-derived joint angle data were visually screened for tracking fidelity. Frames in which the Azure Kinect’s skeletal tracking algorithm failed to maintain anatomical alignment were excluded from analysis. These occurrences were typically characterized by marker drift or disappearance, or abrupt trajectory jumps uncharacteristic of natural movement. No data were excluded solely based on deviation from Kinovea values; if the Kinect markers consistently followed anatomical landmarks throughout the movement, the data were retained. This approach ensured that accuracy evaluation reflected true algorithmic performance rather than artifact-induced error.

### 2.6. Statistical Analysis

Descriptive analysis of the data was performed using means and standard deviations (SD) for continuous variables, and proportions and frequencies for categorical variables. RMSE and intraclass correlation coefficients (ICC3k) with 95% confidence intervals were calculated to assess the accuracy and reliability of the MARS compared to Kinovea. The ICC3k (two-way mixed effects, consistency, average measures) model was selected because the analysis involved a fixed set of measurement systems (MARS and Kinovea) rather than random raters, and the main focus was on the consistency of measurements rather than absolute agreement. MARS joint angles were stratified into clinically meaningful ranges (<45°, 45–89°, 90–134°, and >135°), and RMSE was calculated for each range using the corresponding Kinovea values.

A one-sample *t*-test was conducted to determine if the mean absolute difference (MAD) between MARS and Kinovea was significantly less than 8°. A threshold of ≤8° RMSE was selected as an acceptable clinical margin of error for joint ROM measurements, consistent with previous validation studies of IMU-based and markerless motion capture systems that generally consider error tolerances of up to 10° as clinically significant [9]. All statistical analyses were performed using IBM SPSS Statistics version 27.

## 3. Results

### 3.1. Root Mean Squared Error

The RMSE values for joint angle measurements comparing the MARS and Kinovea systems are presented in Table 2. The lowest RMSE was observed for shoulder abduction (3.96°). In contrast, the highest RMSE was observed for sagittal elbow flexion (11.2°). RMSE values for frontal elbow flexion and shoulder flexion were 5.92° and 9.21°, respectively.

With an overall RMSE of 7.97°, our findings confirm that MARS meets the predefined accuracy threshold of ≤8°.

### 3.2. Intraclass Correlation Coefficient (ICC3k)

The reliability of joint angle measurements between the MARS and Kinovea systems was evaluated using ICC3k with corresponding 95% confidence intervals. Across all joint movements, the ICC3k values demonstrated excellent reliability, ranging from 0.977 to 0.998 (Table 3). The overall ICC3k across all movements was 0.993 (95%CI: 0.992, 0.995), confirming strong agreement between the two systems.

### 3.3. Mean Absolute Difference (MAD)

The MAD between the MARS and Kinovea systems were analyzed to determine whether they were statistically less than 8° using one-sample *t*-test (Table 4).

All joint movements, except sagittal elbow flexion, demonstrated statistically significant MAD values below the 8° threshold.

The total MAD across all joint movements was 4.19° (SD = 5.15), which was significantly less than the 8° threshold, *t*(4937) = −52.04, *p* = 0.000.

### 3.4. Stratified Root Mean Square Error (RMSE)

RMSE was calculated for joint angle measurements by stratifying the MARS joint angles into clinically meaningful ranges (<45°, 45–89°, 90–134°, and >135°) and comparing them to the corresponding Kinovea values (Table 5).

Across all MARS joint angle ranges, shoulder flexion and sagittal elbow flexion consistently demonstrated higher RMSE values compared to shoulder abduction and frontal elbow flexion.

## 4. Discussion

The purpose of this study was to evaluate the feasibility and accuracy of the Microsoft Azure Kinect-powered MARS platform, compared to the widely used Kinovea software for ROM assessment in clinical and pediatric settings. Results demonstrated that MARS achieved excellent reliability, with an overall ICC3k of 0.993 across all movements, with RMSE values below the predefined threshold of 8° for most measures, except sagittal plane elbow flexion.

Stratification of joint movements into clinically meaningful ranges revealed variability in measurement accuracy across different joint angles and planes. Frontal plane movements demonstrated greater accuracy at smaller joint angles, particularly in shoulder abduction, where measurement consistency improved with reduced ROM. Conversely, sagittal plane movements showed less predictable patterns, with the highest RMSE observed during sagittal elbow flexion. These findings highlight potential limitations of MARS, particularly in scenarios where limb positioning may interfere with the depth sensor’s accuracy.

The high ICC3k values observed underscore MARS’ potential for reliable ROM assessment, particularly in settings where traditional tools like handheld goniometers or Kinovea face practical challenges. While Kinovea has been validated in prior research [4,6,10,11], it is not considered a gold-standard motion capture system and is subject to known limitations. Its reliance on manual input introduces operator-dependent variability, and its 2D framework restricts multiplanar analysis and depth perception, thus limiting scalability and objectivity in real-world clinical environments. The Azure Kinect-powered MARS addresses these constraints by automating data collection and analysis through a markerless system, making it well-suited for pediatric populations, where limited cooperation and variable movement patterns can complicate assessment [12,13].

Clinically, MARS offers the advantage of assessing functional, task-based movements rather than isolated ROM, providing a more comprehensive understanding of motor function. This capability allows clinicians to identify compensatory strategies and address movement deficits in a targeted manner, enabling a greater translation of clinical care towards ADL.

A notable distinction between MARS and Kinovea lies in the evaluation time. MARS performs automated, frame-by-frame ROM calculations in real time during the movement sequence, requiring no post-processing or manual input. In contrast, each Kinovea analysis required a 30 min time commitment, requiring manual marker placement and angle calculation at predefined joint positions across each frame. Given the demonstrated agreement with Kinovea, the instantaneous output of MARS reinforces its suitability for clinical implementation, offering a time-efficient yet methodologically robust solution for range of motion assessment without compromising measurement precision.

The comparatively high RMSE values for shoulder flexion and sagittal elbow flexion suggest limitations in MARS’ tracking accuracy during specific movement types. These inaccuracies highlight the methodological challenges of a system constrained by field of view in the form of marker occlusion. When the camera’s view is obstructed by factors such as limbs, long hair, clothing, or overlapping anatomical segments, this can result in a temporary loss of misplacement of markers used for angular calculation. For example, during sagittal plane elbow flexion, flexing the wrist toward the shoulder occasionally caused the system to lose track of the elbow and/or shoulder markers. This resulted in marker misplacement or disappearance and, consequently, inaccuracies in joint angle calculations. These findings underscore a key limitation of markerless systems: reduced reliability in movements involving high flexion angles or overlapping body segments due to marker occlusion. Future iterations of MARS may benefit from advanced algorithmic refinement, to improve accuracy in these scenarios.

Another noteworthy consideration is the reduced accuracy observed in sagittal elbow flexion, particularly at higher ROM ranges. Stratified RMSE values revealed a clear trend of increasing error at more extreme joint angles, with the highest RMSE (15.82°) occurring in sagittal elbow flexion beyond 135°. These deviations may influence clinical decisions in rehabilitation scenarios requiring precise end-range assessment, such as post-operative elbow flexion recovery or orthopedic follow-up. While MARS performed reliably in most ROM domains, the error in this movement suggests a need for caution when interpreting high-flexion sagittal elbow data in isolation.

Additionally, this study’s reliance on Kinovea as the comparative standard introduces a potential limitation, as Kinovea is not considered the gold standard for motion capture. Validation against higher-fidelity systems, such as optical motion capture, would provide a more robust evaluation of MARS’ accuracy. Furthermore, the exclusive focus on a pediatric cohort limits the generalizability of findings to other populations. Expanding future studies to include diverse age groups and clinical settings will be critical for evaluating the broader applicability of MARS in clinical practice. Furthermore, although RMSE provides a useful summary of average measurement error, it may underestimate localized inaccuracies or occasional deviations, particularly in studies with moderate sample sizes and stratified comparisons.

Future research should aim to address these limitations by including clinical populations across a broader range of age groups and functional abilities, as well as expanding the scope of assessment to include lower extremity assessments and more functional tasks such as gait analysis, balance assessment and dynamic movement evaluations. Continued development of MARS will depend on improvements in sensor algorithms, hardware design, and potential integration of artificial intelligence to enhance its utility across a variety of clinical environments and movement settings.

## 5. Conclusions

This study demonstrates that MARS offers a reliable and accurate alternative to traditional ROM assessment tools in pediatric clinical settings. MARS met or exceeded predefined benchmarks for reliability (ICC3k ≥ 0.9) and accuracy (RMSE ≤ 8°) in nearly all movement planes, supporting its validity as a markerless, low-cost, and scalable solution. Unlike conventional methods, MARS enables functional, real-time assessments without the need for wearable sensors or manual ROM analysis, addressing key barriers in pediatric rehabilitation environments. This represents a novel contribution to clinical movement assessment, providing a foundation for broader integration into clinical workflows.

That said, several constraints must be acknowledged. Elevated RMSE in shoulder flexion and sagittal elbow flexion highlights the need for improved occlusion handling and algorithmic refinement. Additionally, findings are based on a pediatric cohort and should not be generalized without further validation across age groups and clinical populations. Future work should expand validation against gold-standard systems and explore its application across diverse age groups and movement contexts.

## Figures and Tables

**Figure 1 sensors-25-04269-f001:**
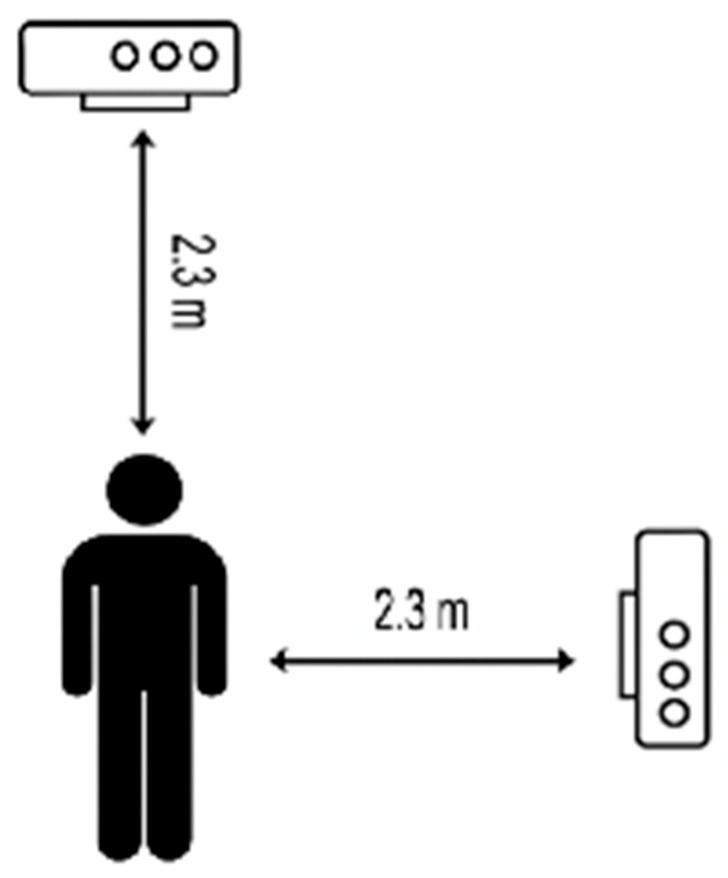
Schematic representation of Azure Kinect camera placement for ROM assessment.

**Figure 2 sensors-25-04269-f002:**
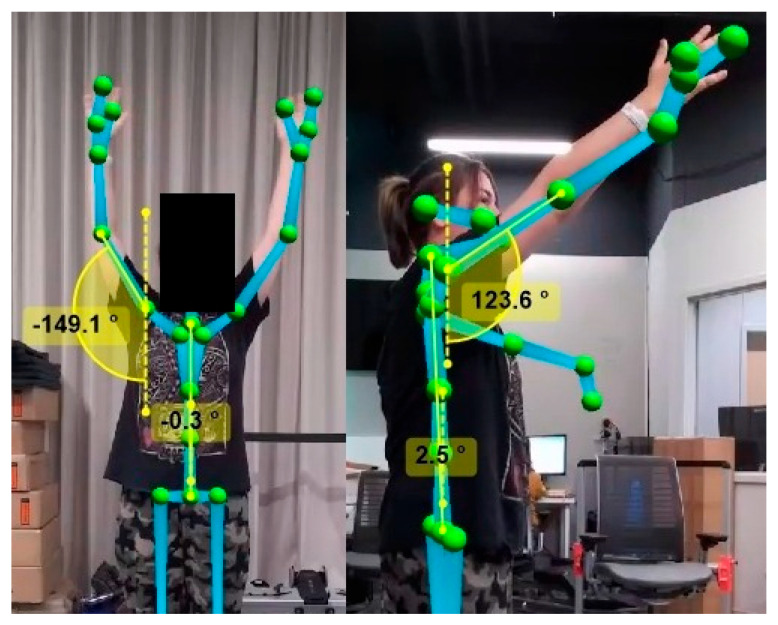
Measurement of shoulder range of motion (ROM) with torso compensation using Kinovea (Version 0.9.5). Torso deviation was quantified by calculating the angle between pelvic and neck markers in both the frontal and sagittal planes. Shoulder ROM (flexion and abduction) was then determined using vertical angle measurements between markers placed on the shoulder and elbow joints.

**Figure 3 sensors-25-04269-f003:**
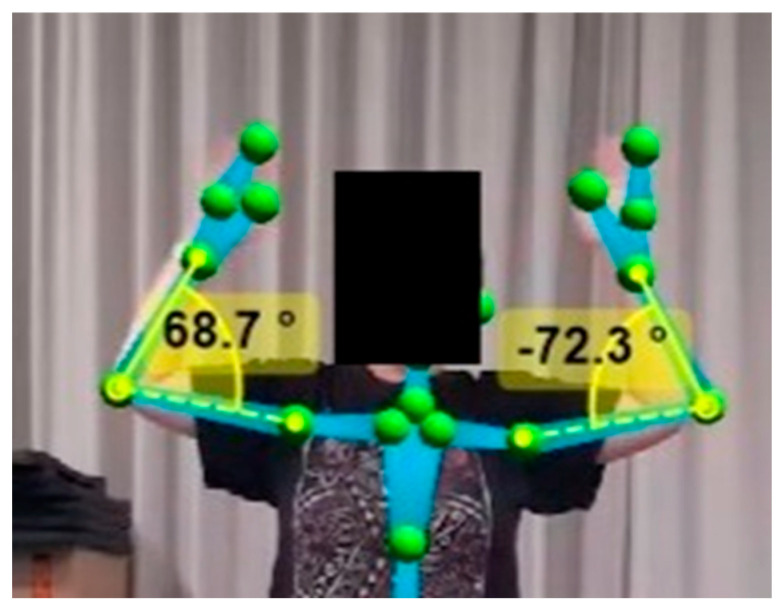
Elbow joint range of motion (ROM) measurement in the frontal plane using Kinovea (yellow) as the validation tool.

**Table 1 sensors-25-04269-t001:** Participant Demographics (n = 65).

Variable	Mean (SD) or n (%)
Age (years)	12.7 (3.3)
Height (cm)	154.5 (15.9)
Weight (kg)	53.2 (20.2)
BMI	21.7 (6.2)
Gender	
Male	29 (44.6%)
Female	36 (55.4%)
Dominant Arm	
Right	60 (92.3%)
Left	5 (7.7%)
Race	
American Indian or Alaska Native	1 (1.5%)
Asian	5 (7.7%)
Black	2 (3.1%)
Black or African American	3 (4.6%)
More than one race	7 (10.8%)
Unknown or not reported	1 (1.5%)
White	46 (70.8%)
Ethnicity	
Hispanic or Latino	10 (15.4%)
Not Hispanic or Latino	49 (75.4%)
Unknown or not reported	6 (9.2%)

**Table 2 sensors-25-04269-t002:** Root Mean Squared Error (RMSE) values for Joint Angle Measurements Between MARS and Kinovea Systems.

Joint Angle	RMSE (°)
Shoulder ABD	3.96
Frontal Elbow FLX	5.92
Shoulder FLX	9.21
Sag. Elbow FLX	11.20
Total	7.97

**Table 3 sensors-25-04269-t003:** Intraclass Correlation Coefficient (ICC3k) for Joint Angle Measurements Between MARS and Kinovea Systems.

Joint Angle	ICC3k	95% CI
Shoulder ABD	0.998	0.997, 0.998
Frontal Elbow FLX	0.992	0.978, 0.996
Shoulder FLX	0.989	0.987, 0.991
Sag. Elbow FLX	0.977	0.971, 0.982
Total	0.993	0.992, 0.995

**Table 4 sensors-25-04269-t004:** Mean Absolute Differences (MAD) and One Sample t-Test Results Evaluating System Accuracy Below 8 Degrees.

Joint Angle	n	MAD (°)	SD (°)	*t*-Statistic	*p*-Value
Shoulder ABD	2283	2.51	3.06	−85.77	0.000
Frontal Elbow FLX	1394	4.15	4.23	−33.98	<0.001
Shoulder FLX	627	6.57	6.45	−5.54	<0.001
Sag. Elbow FLX	634	7.96	7.89	−0.12	0.904
Total	4938	4.19	5.15	−52.04	0.000

**Table 5 sensors-25-04269-t005:** RMSE for Stratified MARS Joint Angles Using Corresponding Kinovea Measurements.

MARS Joint Angle	Shoulder ABD		Frontal Elbow FLX		Shoulder FLX		Sag. Elbow FLX		Total	
	n	RMSE	n	RMSE	n	RMSE	n	RMSE	n	RMSE
<45	293	2.81	310	4.45	54	10.90	252	9.86	909	6.58
45–89	740	2.65	602	5.16	155	8.19	129	13.49	1626	5.82
90–134	790	3.44	415	7.57	129	11.79	231	10.63	1565	7.02
>135	460	6.40	67	6.66	289	7.98	22	15.82	838	7.39

## Data Availability

The data and software used in this study are proprietary and are not publicly available due to intellectual property restrictions. Interested parties may contact the corresponding author for limited access, subject to approval.

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
