# Peer review of "Assessing the Accuracy and Reliability of the Monitored Augmented Rehabilitation System for Measuring Shoulder and Elbow Range of Motion"

_sensors, 2025, doi:10.3390/s25144269_

Round 1

Reviewer 1 Report

Comments and Suggestions for Authors

This study aimed to analyze the usefulness and precision of the Microsoft Azure Kinec MARS platform in comparison to the commonly utilized Kinovea software for range of motion evaluation in clinical and pediatric environments. The results indicated that MARS exhibited exceptional accuracy and reliability across all motions, with RMSE values below the established threshold of 8° for the majority of measurements, except for sagittal plane elbow flexion.

The introduction provides sufficient background and includes most of the relevant references. However, in Line 113, the authors stated, “Frontal plane movements were analyzed using the video from the front camera with the joint angle calculation feature of Kinovea (Version 0.9.5. Charmant, J. & contributors, http://www.kinovea.org).” Sagittal plane movements were determined using the angles reported by the front camera and were subsequently validated by the second side camera. I believe they need to explain and elucidate more about the rationale for utilizing this approach.

I couldn’t find a sample size calculation in the methodology section. It is unclear whether 65 participants can provide sufficient statistical power to test the hypothesis. If this study is based on preliminary findings, it requires further clarification. Additionally, the sample exhibits a significant age range, ranging from 5 to 18 years. This study also requires further explanation and clarification.

The results back up the conclusions. However, since the discussion spans less than one page, it requires additional content. It may be beneficial to add a paragraph discussing the advantages of the Microsoft Azure Kinect-powered MARS platform in comparison with other existing markerless systems. Furthermore, it will be necessary to discuss the advantages and limitations of this proposed method. They should also talk about this method's future uses in research settings like Swam Htet's study, which is linked below, as well as for other non-clinical or pediatric environments, for example, sport-related research applications. 

https://bmcresnotes.biomedcentral.com/articles/10.1186/s13104-023-06617-3

Reviewer 2 Report

Comments and Suggestions for Authors

The study investigates the feasibility of a Microsoft Kinect-based system compared to Kinovea for pediatric range of motion (ROM) assessment. This review evaluates the manuscript's structure, clarity, methodology, and interpretation, providing critical feedback organized by theme and referencing line numbers.

(Lines 30–73)

The introduction effectively contextualizes the clinical need for accessible and reliable ROM measurement tools, particularly in pediatric settings. However, the rationale for selecting Kinovea as the comparator system could be more explicitly tied to limitations of existing gold-standard systems such as Vicon. Moreover, the potential limitations of Kinovea—manual annotation, 2D constraints—should be briefly acknowledged upfront to prepare readers for the comparative analysis. Moreover, the statement at lines 49-50 (“Handheld goniometry remains the primary tool for ROM in clinical settings due to 49 its simplicity and accessibility.”) is misleading as digital goniometry (especially those based on inertial sensors) have becoming increasingly popular both for commercial use and for research purposes and their use has been validated in the scientific literature as systems for tracking rehabilitation outcomes in ambulatory settings (please acknowledge, for instance, the contributions of

  • Tranquilli C, et al (2013). Ambulatory joint mobility and muscle strength assessment during rehabilitation using a single wearable inertial sensor. MEDICINA DELLO SPORT, 66(4):683-97.
  • Parel I, et al (2023) Shoulder Rehabilitation Exercises With Kinematic Biofeedback After Arthroscopic Rotator Cuff Repair: Protocol for a New Integrated Rehabilitation Program. JMIR Res Protoc;12:e35757

The “filling the gap” that can strengthen your work in light of what I’ve just stated is that the abovementioned studies (based on inertial sensors) are poorly suited to children….much less than a markerless system like the Azure (line 56-58).

(Lines 74–129)

The methodology section is overall well-structured, but several technical details need expansion. For instance, the description of Azure Kinect placement (Line 88) would benefit from a diagram or schematic to illustrate camera alignment and body tracking coverage. Likewise, the selection of anatomical landmarks for joint angle calculation (Line 98) should include justification based on biomechanical principles or prior studies. Additionally, the inclusion of torso compensation techniques is commendable (Lines 123–129), though the authors should elaborate on whether compensation was applied algorithmically or post-hoc.

(Lines 143–152)

The statistical methods are appropriate and clearly explained. However, the rationale for the chosen RMSE threshold (≤ 8°) should be cited. Is this threshold derived from clinical significance, existing validation standards, or prior publications? Clarifying this would help contextualize the findings. Furthermore, although ICC3k is a robust measure for reliability, the manuscript could briefly explain why this particular ICC model was selected over others (e.g., ICC(2,1) or ICC(A,1)).

(Lines 154–224)

The results are well-presented and easy to interpret. That said, the stratified RMSE values reveal important nuances (e.g., higher error at extreme ROMs), which could be highlighted more explicitly in both the results and discussion sections. The relatively poor accuracy in sagittal elbow flexion (Line 183, 219) deserves greater emphasis, particularly as this could impact clinical decision-making in certain rehabilitation scenarios. The excellent ICC values are impressive but should be balanced against the known limitations of RMSE sensitivity in smaller datasets.

(Lines 225–276)

The discussion thoughtfully integrates study findings with practical applications. However, the comparison to Kinovea (Lines 242–244) is a bit one-sided. Since Kinovea is not a gold-standard system, its limitations should be critically discussed to ensure the study’s conclusions about MARS are not overstated. Additionally, the limitations paragraph (Lines 253–269) is appropriately included but could benefit from stronger language about the potential risks of overgeneralizing from pediatric data alone. For example, how might findings differ in adult or geriatric cohorts?

(Lines 278–285)

The conclusion appropriately reiterates the study’s strengths. That said, a more critical stance acknowledging current constraints (e.g., sensor occlusion, sagittal errors) would demonstrate greater scientific maturity. Moreover, the manuscript would benefit from a brief mention of regulatory, ethical, or commercialization aspects, especially given the disclosure of potential conflict of interest (Lines 307–308).

Reviewer 3 Report

Comments and Suggestions for Authors
  1. The paper fails to articulate a clear ​novel contribution  or quantify ​significance. Explicitly position the innovation in the Introduction conclusion or Methodology opening section.
  2. Critical parameters (sampling rate in Hz & ​resolution in pixels) for Azure Kinect are unreported. ​Comparative device evaluations require identical parameter baselines for validity. 

  3. The 45° starting angle in Table 5 lacks biomechanical justification (neutral position/0° is conventional). Cite supporting literature or human kinematics evidence for this threshold selection.

  4. Non-standard abbreviations violate conventions. ​Define all acronyms at first use. Add a glossary if acronyms exceed 10 items.

  5. Excessively high RMSD errors  for Shoulder FLX and Sagittal Elbow FLX (Table 2) indicate methodological issues. Investigate causes: marker occlusion? Algorithmic compensation failure? Discuss limitations.

  6. The ​outlier handling protocol for artifacts is absent. Detail data-cleaning methods.

  7. Accuracy claims assume consistent evaluation time, but duration differences between MARS and Kinovea may distort comparisons. Report time costs and discuss time-accuracy trade-offs.

Round 2

Reviewer 1 Report

Comments and Suggestions for Authors

The authors address all the suggested points. There are no additional comments from my side.

Reviewer 2 Report

Comments and Suggestions for Authors

thanks for being amenable to my suggestions. I think the paper has improved and now elegible for publication 

Reviewer 3 Report

Comments and Suggestions for Authors

The author has answered all my questions, and I have no further suggestions for this paper.